# Diagnostics of *BAP1*-Tumor Predisposition Syndrome by a Multitesting Approach: A Ten-Year-Long Experience

**DOI:** 10.3390/diagnostics12071710

**Published:** 2022-07-13

**Authors:** Marika Sculco, Marta La Vecchia, Anna Aspesi, Michela Giulia Clavenna, Michela Salvo, Giulia Borgonovi, Alessandra Pittaro, Gianluca Witel, Francesca Napoli, Angela Listì, Federica Grosso, Roberta Libener, Antonio Maconi, Ottavio Rena, Renzo Boldorini, Daniela Giachino, Paolo Bironzo, Antonella Maffè, Greta Alì, Lisa Elefanti, Chiara Menin, Luisella Righi, Cristian Tampieri, Giorgio Vittorio Scagliotti, Caterina Dianzani, Daniela Ferrante, Enrica Migliore, Corrado Magnani, Dario Mirabelli, Giuseppe Matullo, Irma Dianzani

**Affiliations:** 1Department of Health Sciences, Università del Piemonte Orientale, 28100 Novara, Italy; marika.sculco@uniupo.it (M.S.); marta.lavecchia@uniupo.it (M.L.V.); anna.aspesi@med.uniupo.it (A.A.); michela.clavenna@uniupo.it (M.G.C.); michela.salvo@uniupo.it (M.S.); 20020080@studenti.uniupo.it (G.B.); renzo.boldorini@med.uniupo.it (R.B.); 2Unit of Pathology, AOU Maggiore Della Carità Hospital, 28100 Novara, Italy; 3Pathology Unit, AOU Città della Salute e della Scienza, 10126 Turin, Italy; pittaro.alessandra@gmail.com; 4Department of Medical Sciences, Università di Torino, 10126 Turin, Italy; gianlucarobert.witel@unito.it (G.W.); giuseppe.matullo@unito.it (G.M.); 5Department of Oncology, Università di Torino at San Luigi Hospital, 10043 Turin, Italy; francesca.napoli@unito.it (F.N.); alisti@live.it (A.L.); paolo.bironzo@unito.it (P.B.); luisella.righi@unito.it (L.R.); giorgio.scagliotti@unito.it (G.V.S.); 6Mesothelioma Unit, AO SS. Antonio e Biagio e Cesare Arrigo, 15121 Alessandria, Italy; federica.grosso@ospedale.al.it; 7Department of Integrated Activities Research and Innovation, AO SS. Antonio e Biagio e Cesare Arrigo, 15121 Alessandria, Italy; rlibener@ospedale.al.it (R.L.); amaconi@ospedale.al.it (A.M.); 8Thoracic Surgery Unit, AOU Maggiore della Carità, 28100 Novara, Italy; ottavio.rena@uniupo.it; 9Medical Genetics Unit, Department of Clinical and Biological Sciences, Università di Torino, AOU S. Luigi Gonzaga, 10043 Turin, Italy; daniela.giachino@unito.it; 10Genetics and Molecular Biology Unit, Santa Croce e Carle Hospital, 12100 Cuneo, Italy; maffe.a@ospedale.cuneo.it; 11Unit of Pathological Anatomy, University Hospital of Pisa, 56126 Pisa, Italy; greta.ali@gmail.com; 12Immunology and Diagnostics Molecular Oncology Unit, Veneto Institute of Oncology IOV-IRCCS, 35128 Padua, Italy; lisa.elefanti@iov.veneto.it (L.E.); chiara.menin@iov.veneto.it (C.M.); 13Pathology Unit, Department of Medical Sciences, AOU Città della Salute e della Scienza, 10126 Turin, Italy; cristian.tampieri@unito.it; 14Department of Plastic, Reconstructive and Cosmetic Surgery, Campus Bio-Medico University Hospital, 00128 Rome, Italy; c.dianzani@unicampus.it; 15Unit of Medical Statistics, Department of Translational Medicine, Università del Piemonte Orientale and Cancer Epidemiology, CPO Piemonte, 28100 Novara, Italy; daniela.ferrante@med.uniupo.it (D.F.); corrado.magnani@med.uniupo.it (C.M.); 16Unit of Cancer Epidemiology, Città della Salute e della Scienza, University-Hospital and Center for Cancer Prevention (CPO), 10126 Turin, Italy; enrica.migliore@cpo.it (E.M.); dario.mirabelli@gmail.com (D.M.); 17Interdepartmental Center for Studies on Asbestos and other Toxic Particulates “G. Scansetti”, Università di Torino, 10126 Turin, Italy

**Keywords:** *BAP1*-TPDS, mesothelioma, melanoma, germline variants, cancer genome, diagnostics, immunohistochemistry

## Abstract

Germline mutations in the tumor suppressor gene BRCA1-associated protein-1 (*BAP1*) lead to *BAP1* tumor predisposition syndrome (*BAP1*-TPDS), characterized by high susceptibility to several tumor types, chiefly melanoma, mesothelioma, renal cell carcinoma, and basal cell carcinoma. Here, we present the results of our ten-year experience in the molecular diagnosis of *BAP1*-TPDS, along with a clinical update and cascade genetic testing of previously reported *BAP1*-TPDS patients and their relatives. Specifically, we sequenced germline DNA samples from 101 individuals with suspected *BAP1*-TPDS and validated pathogenic variants (PVs) by assessing *BAP1* somatic loss in matching tumor specimens. Overall, we identified seven patients (7/101, 6.9%) carrying six different germline *BAP1* PVs, including one novel variant. Consistently, cascade testing revealed a total of seven *BAP1* PV carriers. In addition, we explored the mutational burden of *BAP1*-TPDS tumors by targeted next-generation sequencing. Lastly, we found that certain tumors present in PV carriers retain a wild-type *BAP1* allele, suggesting a sporadic origin of these tumors or a functional role of heterozygous *BAP1* in neoplastic development. Altogether, our findings have important clinical implications for therapeutic response of *BAP1*-TPDS patients.

## 1. Introduction

The BRCA1-associated protein-1 (*BAP1*) gene, composed of 17 exons located on chromosome 3p21, encodes for a ubiquitin carboxy-terminal hydrolase that modulates multiple cellular activities, such as transcription control, DNA repair, chromatin modification, mitochondrial function, and cell death [1].

Carriers of heterozygous germline pathogenic variants (PVs) of *BAP1* are affected by a hereditary condition called *BAP1* tumor predisposition syndrome (*BAP1*-TPDS), characterized by predisposition to a wide range of tumors, whose core spectrum includes mesothelioma, cutaneous and uveal melanoma (CM and UM, respectively), renal cell carcinoma (RCC), and melanocytic *BAP1*-mutated atypical intradermal tumor (MBAIT). More recently, other tumors, such as basal cell carcinoma (BCC), cholangiocarcinoma, and meningioma, have been associated with *BAP1*-TPDS, albeit characterized by a much lower incidence [2,3,4]. Tumors not yet officially included in the *BAP1*-TPDS spectrum, but found in carriers of *BAP1* PVs, are breast cancer, non-small cell lung adenocarcinoma, and neuroendocrine carcinoma [5,6,7].

*BAP1*-TPDS is inherited in an autosomal dominant fashion [2,5,8,9,10,11] with a penetrance close to 100% in older age individuals [7,12]. In general, carriers of germline *BAP1* PVs develop tumors at a younger age than that of sporadic patients [4,7,12,13].

Several studies have shown that many—but not all—mesothelioma patients with germline *BAP1*-TPDS are characterized by prolonged survival compared to wild-type (wt) *BAP1* patients [8,9,12,14,15,16,17,18]. In contrast, *BAP1* PV carriers affected by melanoma—both cutaneous and uveal—exhibit increased risk of metastasis, suggesting a shorter survival rate [19]. A similar outcome is also observed in RCC patients with germline *BAP1* mutations [11]. On the other hand, long-term follow-up data are not available for patients with BCC, most likely owing to the fact that these tumors are generally not aggressive [6,20].

Among *BAP1*-TPDS-associated tumors, MBAITs are regarded as benign skin lesions characterized by *BAP1* inactivation. Since they arise during the first two decades of life and tend to increase in number with age, their detection can be useful for early cancer diagnosis as they help identify *BAP1* PV carriers several years before the development of more aggressive tumors belonging to the *BAP1*-TPDS spectrum [6,9,21].

*BAP1*-TPDS individuals may also develop meningiomas, which are primary central nervous system (CNS) tumors of the meninges. Even though these tumors are generally slow-growing, highly aggressive grade III rhabdoid meningiomas can sometimes be found in TPDS patients [22,23].

*BAP1*-TPDS should be suspected if an individual has two or more tumors of the *BAP1*-TPDS spectrum (including MBAITs) or has one *BAP1*-TPDS malignancy and a first- or second-degree relative with a tumor included in the *BAP1*-TPDS spectrum, excluding cases or families with two BCCs and/or CMs because of their high frequency in the general population [6]. In this regard, Walpole and colleagues have recently shown that it is essential to test and identify germline *BAP1* carriers in order to implement surveillance, which ultimately leads to improved survival and cost savings for the healthcare system [4]. Indeed, cascade genetic screening in patients’ relatives carrying germline mutations can be critical for timely initiation of therapies and/or preventive measures (e.g., limiting sunlight exposure) [6].

*BAP1* expression is frequently lost in mesothelioma, UM, and RCC due to somatic inactivation [7,18,24]. More specifically, this inactivation has been observed in 30–60% of sporadic mesotheliomas [7,25,26,27], a characteristic that has been used in diagnostics [28].

Although the tumor genome of sporadic mesothelioma patients has been extensively investigated, the tumor genome of patients affected by *BAP1*-TPDS has yet to be systematically characterized. A single paper analyzed the genome of several metachronous tumors in a *BAP1*-TPDS patient [29]. In a recent study involving 17,152 patients with different tumor types, Srinivasan et al. reported some information of the somatic genome of six patients with TPDS [30].

We report here the results of our 10-year-long clinical diagnostics of *BAP1*-TPDS. Overall, we performed *BAP1* Sanger sequencing and multiplex ligation-dependent probe amplification (MLPA) assay on germline DNA samples from 101 suspected *BAP1*-TPDS individuals. To validate the functional role of the identified variants, we searched for *BAP1* loss in the tumor tissues from the patients harboring mutations. When possible, germline genetic testing and tumor analyses were performed on the patients’ relatives, as well.

The diagnostic and prognostic significance of our integrated analysis of germline and somatic alterations in *BAP1*-TPDS individuals will be discussed in the context of precision medicine therapies and preventive strategies.

## 2. Materials and Methods

### 2.1. Patients

*BAP1*-TPDS individuals were enrolled in the study in the following Italian healthcare settings: Antonio e Biagio e Cesare Arrigo Hospital (Alessandria); Pathology Unit at San Luigi Gonzaga Hospital (Orbassano, Turin); Molecular Genetics and Biology Unit at Santa Croce e Carle Hospital (Cuneo); Division of Dermatology at Maggiore della Carità Hospital (Novara); Genetics and Pathology Units at City of Health and Science Hospital (Turin); Immunology and Diagnostics Molecular Oncology Unit of Veneto Institute of Oncology IOV-IRCCS (Padua).

Patients were selected according to the following criteria: a family history of cancers and the presence of one tumor of the *BAP1*-TPDS spectrum affecting the proband or a family member. The clinical features of the 101 patients are reported in Appendix A. Thirty-nine out of 101 patients with CM or multiple CMs with a family history of melanoma were previously reported [31].

### 2.2. Germline Genome Analyses

Genomic DNA from whole peripheral blood of 98 patients was analyzed by Sanger sequencing of the 17 exons, intron–exon boundaries, and promoter region (~1000 bp upstream of the ATG) of *BAP1* (NM_004656.2) as previously described [32]. The germline DNA extracted from the peripheral blood of patients MM981, MM1012, and MM400 was analyzed by NGS technique using a customized gene panel (kit QIAseq™ Targeted DNA Panels, Qiagen, Hilden, Germany) based on amplicon technology, which targets the exonic regions and the splice junctions of five known familial melanoma susceptibility genes (i.e., *BAP1*, *CDKN2A*, *CDK4*, *POT1*, and *MITF*). Briefly, genomic dsDNAs were subjected to fragmentation, adapter ligation with unique molecular indices, target enrichment, and library amplification according to the manufacturer’s protocol (Qiagen, Hilden, Germany). The library pool was sequenced with the MiSeq platform (Illumina, San Diego, CA, USA) using the reagents of the v3-600 flow-cell (Illumina, San Diego, CA). The FASTQ files were imported into the CLC Genomic Workbench program (Qiagen, Hilden, Germany) and analyzed through a pipeline optimized for the identification of germline variants (Identify QIAseq DNA Germline Variants).

Variants were classified as pathogenic according to the American College of Medical Genetics (ACMG) guidelines [33,34]. The identified PVs were also evaluated on tumor DNA by Sanger sequencing. The germline copy number variation (CNV) was assessed by MLPA assay using SALSA MLPA P417 BAP1 probemix (MRC-Holland, Amsterdam, The Netherlands).

The *MSH6* variant identified by NGS in the tumor samples from proband HO19.01 (II-3) and her brother (II-2) was validated by Sanger sequencing on the germline DNAs of these two subjects. Detailed protocols for amplification, sequencing, and tumor analyses are reported in the Appendix A.

### 2.3. Immunohistochemical and LOH Analyses

To determine the functional role of the variants in carcinogenesis, different analyses were performed on formalin-fixed paraffin-embedded (FFPE) tumor tissues or pleural effusion specimens. Immunohistochemistry (IHC) was performed on FFPE tissue sections using an anti-human BAP1 primary antibody (rabbit monoclonal, clone C-4, Santa Cruz Biotechnology, Inc., Santa Cruz, CA, USA). To determine BAP1 LOH, microsatellite analysis on tumor DNA extracted from FFPE specimens was performed. MLPA assays on DNA extracted from the pleural effusion samples were also performed. MLPA, IHC, and microsatellite analyses were performed as described previously [32,35].

### 2.4. RT-PCR Analysis of Splice Variant Effects

Total RNA was extracted from the peripheral blood mononuclear cells after Ficoll-Paque density gradient centrifugation and reverse-transcribed into cDNA using the High Capacity cDNA Reverse Transcription kit (Thermo Fisher Scientific, Waltham, MA, USA) (HO19.01 case) or, alternatively, extracted by the Maxwell^®^ RSC Whole Blood DNA method (Promega, WI, USA) and transcribed using the SuperScript^®^ III First-Strand Synthesis System (Invitrogen, Thermo Fisher Scientific, CA, USA) (MM400 and MM1012 cases).

The cDNA was amplified using the following primer pairs:-forward *BAP1*-exon 1 (F 5′-ATGAATAAGGGCTGGCTGGAGCT-3′)–reverse *BAP1*-exon 4 (R 5′-CTGGTGGGCAAAGAACATG-3′) for the HO19.01 patient;-forward *BAP1*-exon 5 (F 5′-CCCTGAGTCGCATGAAGGA-3′)–reverse *BAP1*-exon 7 (R 5′-GTAGACCTTCAGCCCATCCA-3′) for the MM400 patient;-forward *BAP1*-exon 8 (F 5′-CGAGGAGTGGACAGACAAG-3′)–reverse *BAP1*-exon 10 (R 5′-ACTTGTTGCTGGCTGACTTG-3′) for the MM1012 patient.

The PCR products were loaded in a 2% agarose gel to detect the transcript alterations. Sanger sequencing was performed directly on PCR products or on abnormal-size products extracted from the gel.

### 2.5. Somatic NGS-Targeted Sequencing

DNA-based NGS analysis was performed on tumor specimens using a multigene panel (Oncomine Comprehensive Assay v3, OCAv3) (Thermo Fisher Scientific, Waltham, MA, USA), which targets 161 genes (including *BAP1*) following the manufacturer’s instructions (for details, see: https://www.thermofisher.com/order/catalog/product/A35805 (accessed on 10 January 2022)). Briefly, the tumor areas were selected by a pathologist (LR) by means of focal assistant dissection at the microscope in order to ensure adequate content of tumor cells. Somatic DNA was extracted using a Maxwell^®^ RSC DNA FFPE Kit (Promega, Madison, WI, USA) following the manufacturer’s instructions. DNA concentration was quantified using Qubit™ dsDNA High Sensitivity Assay kit (Thermo Fisher Scientific, Waltham, MA, USA) on the Qubit fluorometer (Thermo Fisher Scientific, Waltham, MA, USA). NGS analysis was based on Ion AmpliSeq technology, which requires 20 ng of input for high-quality FFPE DNA to interrogate the 161 genes (including *BAP1*). Deamination reaction was conducted using uracil-DNA glycosylase—heat labile (Thermo Fisher Scientific, Waltham, MA, USA). All library preparations for Oncomine Comprehensive Assay v3 (OCAv3) were manually performed according to the manufacturer’s instructions. Multiplex PCR amplification was carried out using a DNA concentration of approximately 20 ng as input for both assays. Libraries were loaded on Ion 540™ Chips using the Ion Chef™ System and sequenced using Ion Torrent GeneStudio™ S5 Prime (all Thermo Fisher Scientific, Waltham, MA, USA).

The data were mapped to the human genome assembly 19, loaded as a standard reference genome into Ion Reporter™ Software (v. 5.16) (Thermo Fisher Scientific, Waltham, MA, USA). Torrent Suite Software (TSS) vs 5.12.2 generates BAM files, which were uploaded on the Ion Reporter (IR) server used for initial automated analysis. The quality of the sequencing reaction was verified by filtering the coverage analysis, mapped reads, mean depth, uniformity, and alignment of the target region according to the established *p*-value set from 0.0 to 1.0 [36,37]. Subsequently, visual inspection of BAMs was carried out by graphic alignment using Ion Reporter™ Genomic Viewer (IRGV) (Thermo Fisher Scientific, Waltham, MA, USA). Variants of unknown significance (VUSs) were evaluated using MetaLR and MetaSVM, which combine 10 in silico prediction tools (i.e., SIFT, PolyPhen-2 HDIV, PolyPhen-2 HVAR, GERP^++^, Mutation Taster, Mutation Assessor, FATHMM, LRT, and SiPhy) [38]. Variants described in ClinVar (www.clinvar.org (accessed on 18 January 2022)) as benign or likely benign were discarded. All samples were sequenced with a mean coverage of 2000×, with ≥95% of uniformity of amplicon target that had at least 500 reads. The variant allele frequency (VAF) was calculated as the percentage of sequence reads observed matching a specific DNA variant divided by the overall coverage at that locus. VAF is thus a surrogate measure of the proportion of DNA molecules in the original specimen carrying the variant. Copy number (CN) value was set at 2 for autosomal amplicons, and copy gains (≥3) or losses (≤1) were detected.

### 2.6. Cascade Testing and Follow-Up

Cascade genetic screening was performed on 9 relatives of the identified PV carriers. For 8 relatives, the search for the specific germline PV was performed by Sanger sequencing on germline DNA, whereas for one (MPM_HO1901 II-2) on the available tumor tissue. The tumor tissue of carriers was also analyzed when available. An update of the clinical and tumor molecular features of three carriers we previously described—i.e., the proband’s daughter in family A (III-2), the proband’s brother in family A1 (II-2), and the proband’s daughter in family A1 (III-2) [31,35]—is reported in Table 1.

## 3. Results

### 3.1. Identification of Families with BAP1-TPDS

Upon sequencing germline DNAs from 101 Italian patients with suspected *BAP1*-TPDS (Appendix A), we found six different germline *BAP1* PVs harbored by patients from seven families (Table 1) (Appendix A). Four of these families (carriers of the PVs c.46_47insA, c.1153C>T p.Arg385*, c.783+2T>C, and c.38-1G>T) had already been described by our group [31,32,35,38]. Table 1 provides an update on the clinical features of these probands and/or their relatives. The three previously unreported families carry the following *BAP1* PVs: c.783+2T>C (the same as ID_5), c.605G>A p.Trp202*, and c.376-2A>G.

Of the six aforementioned PVs, which appear to be very rare in the general population (MAF < 0.0001) [35], two are nonsense, three are splicing, and one is a frameshift mutation.

MLPA analysis of germline DNA did not reveal any deletions or duplications of *BAP1*.

### 3.2. Family A

Family A was one of the first families to be tested for *BAP1* mutations in 2015 [35] due to a known familial history of mesothelioma. Specifically, three members of this family (i.e., the proband, her sister, and her mother) were all affected by mesothelioma—either pleural or peritoneal—and four of them (i.e., the proband, her daughter and son, and the mother of the proband) carried the c.46_47insA PV (Appendix A). The other family members were not tested. As the proband’s daughter (III-2) had been diagnosed with meningioma—in 2017, at the age of 48—we sought to determine whether this tumor also displayed loss of BAP1 protein expression. IHC analysis of FFPE tissue revealed robust BAP1 protein expression in the nuclei of the stromal cells, but it failed to detect BAP1 expression in the tumor cells (Figure 1), suggesting a functional role of the variant in this meningioma.

### 3.3. Family A1

Similarly to family A, family A1 had also already been reported by our group [31]. Proband II-5 presented with the following metachronous tumors: pleural mesothelioma (PlM), multiple melanoma, meningioma, and basal cell carcinoma (BCC), and carried the c.1153C>T p.Arg385* PV. The follow-up showed that her carrier daughter (III-2) had developed a CM in 2015, at the age of 33 [31], whereas her brother (II-2) had developed a UM in 2014 and a PlM in 2019, at 59 and 64 years old, respectively. Upon genome sequencing, II-2 was found to harbor the nonsense c.1153C>T p.Arg385* variant (Appendix A). No other family members were available for *BAP1* testing. IHC analyses performed on the proband’s BCC and on her brother’s UM revealed loss of BAP1 expression (Figure 2). Proband II-5 died in 2019, nine years after being diagnosed with PlM, whereas her brother (II-2) died two years after the same diagnosis. The proband’s daughter (III-2) is still alive.

### 3.4. Family ID_5

Family ID_5 had also been previously described by our group [32]. The proband carries the canonical splice-site variant c.783+2T>C, which has led to a complete loss of BAP1 protein expression, as judged by IHC. This is consistent with MLPA analysis of the proband’s pleural effusion, showing a 50% deletion of *BAP1* [32]. In the same study, we also showed by FISH analysis that loss of BAP1 expression was likely due to different chromosomal alterations, such as homozygous or heterozygous deletion or monosomy. Specifically, FISH analysis of pleural effusion, which contains a high number of normal cells, revealed that BAP1 biallelic loss was 24%, while the same analysis of FFPE showed that the loss was 73% (Table 2) [32].

In addition, we also performed microsatellite analysis on the DNA extracted from pleural effusion of patient ID_5 III-1, recording a 50% reduction in one allele (Table 2 and data not shown). Moreover, we sought to identify specific somatic *BAP1* molecular alterations. NGS analysis of FFPE PlM tissue revealed the already known germline variant (VAF 52.96%, coverage > 500×) but failed to detect any other pathogenic mutations. However, a significant loss (*p* = 0.006) of the gene copy number was recorded, albeit fairly heterogeneous, confirming the previously described heterogeneous loss detected by FISH (Table 2) [32]. This mutation results in exon 9 skipping, as shown by the experiments described below.

### 3.5. Family PD-601

This family has never been described before (Appendix A). The MM1012 female proband had a history of CM, the first one diagnosed at 55 years. Her mother and paternal aunt had lung cancer. A cousin—daughter of one of her maternal aunts—had been diagnosed with CM, while another one—a daughter of another maternal aunt with leukemia—had an unspecified womb cancer. Moreover, her paternal grandmother had suffered from liver cancer.

The proband germline DNA was tested for *BAP1* and revealed the canonical splice-site variant c.783+2T>C, also found in patient ID_5 III-1. Although the melanoma cells showed a weak focal BAP1 nuclear expression at IHC (Appendix A), the DNA extracted from the FFPE tumor specimens revealed LOH in the tumor (Appendix A). Cascade testing for this family was not performed. We tested the effect of the variant on *BAP1* splicing by analyzing the transcript of the proband. cDNA amplification revealed two *BAP1* PCR products of different sizes, consistent with exon 9 skipping (Appendix A). Sequencing of the fragments confirmed exon 9 skipping, supporting a pathogenic role of this variant.

### 3.6. Family ID MPM_HO1901

The proband (II-3) (Appendix A) had developed PlM (in 2020, age 58), RCC (in 2007, age 46), and lung adenocarcinoma (LUAD) (in 2008, age 47) and carried the canonical splice-site variant c.38-1G>T [39].

To demonstrate the effect of this variant on *BAP1* splicing, we amplified the cDNA of patient MPM_HO1901 II-3 by using primers located on exon 1 and 4, and observed, along the wt PCR product, a small fragment consistent with exon 2 skipping (Appendix A). Sequencing of the PCR products revealed two shortened transcripts, one caused by exon 2 skipping and the other resulting from the use of a cryptic acceptor site in exon 2 (Appendix A), that are expected to encode two proteins lacking 10 or eight amino acids, respectively. These mutant proteins may be unstable or may have lost their deubiquitinase activity, which requires the N-terminal of BAP1. Overall, these data are consistent with a likely pathogenic effect of this variant.

Next, we performed IHC on the proband’s tumor tissues to evaluate BAP1 status. While PlM and LUAD cells showed a complete loss of BAP1 protein expression (Figure 3), the nuclei of normal stromal cells expressed high levels of BAP1 protein. In contrast to the other tumors, RCC cells displayed a partially unreactive expression possibly due to poor tissue preservation (Appendix A).

NGS analysis confirmed the presence of the germline variant in all the tumor tissues analyzed, albeit with different VAFs (Table 2, Appendix A). In particular, PlM cells harbored the germline splicing variant with a VAF of 89% (coverage 1000×), which was associated with a significant (*p* = 0.00004) gene copy number loss (0.9 ratio) (Appendix A), consistent with the complete somatic deletion of the wt allele. Accordingly, MLPA on DNA extracted from fresh pleural effusion specimens showed a 50% loss of the entire *BAP1* gene (Appendix A), which was again consistent with the complete loss of the wt sequence.

The LUAD tumor tissue carried the germline *BAP1* variant with a VAF of 21.2% (coverage 500×). We also found three somatic mutations with a low VAF. The DNA extracted from the FFPE specimens was partially degraded, so the results could not be adequately interpreted.

Although *BAP1* loss is not often detected in LUAD [40], we did not detect BAP1 protein expression by IHC analysis. Our results shows that LUAD may be associated with *BAP1*-TPDS, as previously reported [29].

NGS analysis of the RCC sample identified a germline mutation with a VAF of 48.48% (coverage 1500×), associated to the pathogenic SNV p.Trp196* (2% VAF, coverage 1500×, *p*-value = 0.002, COSM1424466) reported in reference databases (Table 2). Hence, this tumor might not have been caused by TPDS, at least at its initiation.

Cascade genetic analyses were feasible for the proband’s brothers and daughters. One of the proband’s brothers (II-2) died of PlM in 1997. Unfortunately, after performing IHC on his FFPE samples, all tissues were unstained—even stromal cells—likely due to poor specimen condition (data not shown). Nonetheless, NGS analysis revealed the presence of the germline *BAP1* variant c.38–1G>A with a VAF of 53.54% (coverage > 500×). The tumor also showed the pathogenic somatic variant p.Trp196* (3.36% VAF, coverage 500×, *p*-value = 0.0004) (Table 2). Interestingly, the same somatic variant was harbored by the proband’s RCC.

Upon somatic NGS analysis of both the proband’s (II-3) and her brother’s (II-2) tumor samples, we identified a second variant: p.Arg1076His in *MSH6* (48% VAF in the proband’s mesothelioma and RCC; 25% VAF in her brother’s mesothelioma) (Table 2). This gene (MIM#600678) is responsible for Lynch syndrome (OMIM#614350). No further driver mutations were detected in any of the other genes tested. We also searched for PVs in the other brother of the proband (II-1), who was found mutated at the germline level in *MSH6* but not in *BAP1*. We also performed mutational analyses on the germline DNA of the proband’s healthy daughters, uncovering that one of them carried the PV in *BAP1* (age 23, III-2), while the other one carried the PV in *MSH6* (age 26, III-1).

To deepen the role of this variant, we analyzed the mesothelioma tissue of proband II-3. Since MSH6 was normally expressed, as judged by IHC, and the mismatch repair (MMR) status was not modified (data not shown), we deemed this variant to be benign. Of note, while VarSome classifies this variant as likely pathogenic, ClinVar reports conflicting interpretations (https://varsome.com/variant/hg19/MSH6%3AR1076H?annotation-mode=germline (accessed on 4 March 2022); https://www.ncbi.nlm.nih.gov/clinvar/variation/186361/?oq=((182140[AlleleID]))&m=NM_000179.3(MSH6):c.3227G%3EA%20(p.Arg1076His) (accessed on 4 March 2022)). However, we cannot rule out a tissue-specific effect in the colonocytes.

### 3.7. Family PD-578

Like family PD-601, this family has never been reported (Appendix A). Proband MM981 had multiple CMs (i.e., two melanomas at 68, one at 73, and one at 77 years old) and prostate cancer at the age of 60. He has three sisters: one is healthy; the second one has pancreatic cancer; the third one was diagnosed with breast cancer at 56 years old and died at the age of 60. Her daughter died of mesothelioma at the age of 56. Moreover, the proband’s paternal uncle developed prostate cancer. In this family, we only tested the proband’s germline DNA, identifying a previously unknown *BAP1* stop-gain variant: c.605G>A p.Trp202*.

IHC of two different melanomas of the proband showed loss of nuclear BAP1 protein expression (Figure 4). Cascade genetic analyses were unfeasible.

### 3.8. Family PD-238

This family has never been described before (Appendix A). The MM400 proband was diagnosed at 52 with bladder cancer and at 58 with UM. Her father and one of her paternal aunts (or: distal relative of fifth degree) had also UM. The other paternal aunt developed a brain tumor, whereas the two paternal uncles were diagnosed with lung and stomach cancer, respectively. We only tested the germline DNA from the proband and found the splice-site variant c.376-2A>G in *BAP1*, which is predicted to cause exon 6 skipping, leading to the formation of a premature stop codon after eight amino acids. This variant was previously reported in an Australian family [7], but, to our knowledge, its effect on splicing has not been functionally evaluated. To investigate the effect of the variant on *BAP1* transcript, we evaluated total RNA extracted from whole blood of the PV carrier. cDNA amplification and sequencing revealed exon 6 skipping (Appendix A), supporting a pathogenic role of this variant. Cascade genetic and tumor analyses were not feasible.

## 4. Discussion

The present study reports the results of a ten-year program of genetic testing of *BAP1*-TPDS individuals performed at a single reference center in Italy. Among our study panel of 101 suspected *BAP1*-TPDS individuals, we successfully identified seven patients carrying six different germline *BAP1* PVs, one of which never described before (Table 1). Importantly, we show that certain tumors present in PV carriers retain a wt *BAP1* allele, which implies a sporadic origin of these tumors or a functional role of heterozygous *BAP1* in cancer development. Besides having important clinical implications for *BAP1*-TPDS patients, our findings highlight a number of important issues that still need to be fully addressed. These are discussed below.

### 4.1. Should Testing Criteria Be More Stringent?

Currently, *BAP1* genetic testing is recommended for patients who develop two or more tumors of the *BAP1*-TPDS spectrum or for those affected by only one of those tumors provided they have a first- or second-degree relative with a confirmed *BAP1*-TPDS tumor [6]. Individuals or families with two BCCs and/or CMs should be excluded from *BAP1* genetic testing due to the high frequency of these tumors in the general population [6,11]. On the other hand, families with multiple melanomas should first be checked for the presence of the most common high-penetrance melanoma predisposing genes or subjected to NGS panel analysis. In our study, we adopted less stringent criteria to select patients for germline *BAP1* mutation analysis. We feel that less stringent criteria allowed us to identify *BAP1* carriers from non-typical TPDS families, when family history is incomplete (e.g., ID_5 III-1).

Among our study panel, we found a germline *BAP1* PV prevalence of 6.9% (7/101) (CI95% (2.8–13.8)), which is in agreement with that reported by other studies for familial cases (7.7% (3/39) [32], 6% (9/150) [41]). In sporadic patients with mesothelioma, the prevalence of *BAP1*-TPDS is about 1% [6], which is higher than what we found in our previous study performed on patients highly exposed to asbestos (<0.001%, [35]). For familial melanomas, previous reports showed a prevalence of germline *BAP1* PVs ranging from 20–30% for UM [11,42] to <1% for CM [43,44,45].

### 4.2. The Importance of Detecting Secondary Somatic Mutations of BAP1 in Tumor Tissues

Emerging evidence from prospective clinical trials on *BAP1*-TPDS patients suggests that an early diagnosis of TPDS may be crucial for shaping the personalized therapeutic option offered to these patients [46]. In this regard, Srinivasan et al. [30] have recently proposed that the identification of biallelic mutations of genes involved in DNA repair in cancer tissues, such as *BAP1*, should be routinely performed for all patients so as to prospectively evaluate the effectiveness of personalized therapies. Noteworthily, this study showed that 27% of the tumors arising in individuals with inherited cancer syndromes did not display somatic loss of the second allele, which is a necessary tumor-promoting event according to Knudson’s theory. As a partial explanation for this phenomenon, the authors hypothesize a tissue-specific role of the tumor suppressor or a pathogenic effect of the mutation even when it is in its heterozygous form. Alternatively, they also propose that the bioinformatic technique they used to analyze the tumor might not have been sensitive enough to detect all the mutational alterations [30]. Overall, the study shows that the rate of somatic biallelic inactivation of *BAP1* in tumors developed in carriers of germline PVs is 80–90%, and that the second hit is mostly acquired through LOH [30].

### 4.3. Which Is the Best Strategy to Detect a Secondary Somatic Mutation?

Among the methods used to address this issue, FISH is particularly useful for detecting chromosomal deletions, whereas CNV analysis of NGS sequencing data or MLPA, which can only be performed using fresh tissues, is instrumental for revealing both short and whole gene deletions. In contrast, microsatellite-based analysis is only effective in identifying large deletions. The loss of the second allele of the gene of interest may be carried throughout the entire specimen, representing the major tumor clone—see mesothelioma of proband MPM_HO1901 in this study—or it may only be present in a minor clone—as in the case of the RCC and LUAD of the same proband. Thus, the fact that we cannot rule out complete biallelic loss of a gene of interest, such as *BAP1*, due to technical limitations and/or tumor heterogeneity, raises the possibility that, in the same patient, some tumors may be responsive to treatments designed to kill cells lacking *BAP1*, whereas others might only be partially responsive or totally unresponsive to such treatment.

### 4.4. Tumor Genome in BAP1-TPDS

Analysis of *BAP1*-TPDS tumor genomes has never been thoroughly performed.

In 2020, Shinozaki-Ushiku and colleagues reported the genome analyses of several metachronous tumors in a patient with TPDS [29]. They found loss of BAP1 protein expression in all the tested tumors (6/7). Somatic loss of *BAP1* was due to different mutations in PlM of the right thoracic cavity, peritoneal mesothelioma, lung adenocarcinoma, and bladder cancer, whereas no *BAP1* somatic mutation was observed in cholangiocarcinoma and PlM of the left thoracic cavity. Moreover, they identified a low mutational burden.

Srinivasan et al. have recently shown that tumors from germline PV carriers display fewer driver events than those observed in non-carriers [30]. Fittingly, none of our tumor samples carried somatic alterations in other cancer driver genes besides *BAP1*. However, a limitation of our analysis is that we did not perform a whole-genome study or use techniques able to detect complex rearrangements.

### 4.5. Redefining the Pathogenicity of a Splice Mutation

Another controversial issue in the characterization of *BAP1*-TPDS—and any other genetic disease—is the definition of the pathogenicity of a certain mutation. An intriguing case in this regard is that represented by mutation c.783+2T>C (III-1 in family ID_5 and III-5 in family PD-601). This mutation affects the second base (T) of the canonical GT splice site donor of *BAP1* intron 9. Even though this mutation should in theory be pathogenic because it leads to skipping of *BAP1* exon 9, it has been recently reclassified as a variant of unknown significance (VUS) by Goldberg and co-workers [47]. However, this reclassification may have been biased by the use of a forward primer mapping on exon 9—which is therefore not suitable to identify exon 9 skipping—when sequencing *BAP1* transcript from tumor specimens. Indeed, when we repeated the same experiment, but this time using a forward primer designed on exon 8, we were able to detect an alternative *BAP1* splicing consistent with exon 9 skipping, supporting a pathogenic role of the c.783+2T>C variant. Interestingly, it has been estimated that GT>GC substitution within the canonical 5′ splice site can sometimes be partially tolerated, leading to variable amounts of canonical transcripts (1–84%) [48]. Thus, it is possible that this abnormal splicing does not occur in 100% of mutated transcripts. It should be nevertheless pointed out that we eventually observed LOH in the tumor tissue.

Taken all together, these data indicate that the variant c.783+2T>C leads to altered *BAP1* transcript, but given that this effect may be incomplete, it should be regarded as likely pathogenic.

### 4.6. Cascade Genetic Testing

Guidelines published in Gene Reviews [6] recommend that carriers of *BAP1* PVs should undergo surveillance for *BAP1*-TPDS tumors. Indeed, patients who are affected by one *BAP1*-related malignancy are at increased risk of developing other cancers belonging to the *BAP1*-TPDS spectrum, and early tumor detection may allow a more effective management of such tumors [4]. For example, dermatologic screening, combined with preventive measures (e.g., limiting sun exposure), may lower the risk of *BAP1*-TPDS patients developing severe forms of CM. Likewise, ocular screening for UM should be highly recommended.

With regard to mesothelioma, there is no consensus on the most effective screening modalities. Before recommending chest CT to asymptomatic individuals with previous asbestos exposure, the subsequent increased risk of cancer due to radiation exposure should be evaluated [6]. We have previously shown that the combined risk of genetic predisposition and asbestos exposure results in an increased risk of mesothelioma in the carriers of germline mutations [39]. The identification of *BAP1* mutation carriers with mesothelioma can be extremely useful because these patients may potentially benefit from precision medicine, as shown for MPM_HO1901 [14,17,39,46].

In our cohort of *BAP1* PV carrier families, we identified mutations in four relatives who were healthy at the genetic testing. On a recent follow-up, we found that three of them had been diagnosed with confirmed tumors of the *BAP1*-TPDS spectrum, while only one, the 23-year-old daughter of MPM_HO1901, was still healthy. Nevertheless, screening measures have been implemented for this subject.

Regarding associations between specific *BAP1* mutations and a peculiar disease phenotype, in 2018 a comprehensive study [7] collating data from 181 *BAP1*-TPDS families showed that among patients who developed mesothelioma, melanoma, and other tumors, those carrying a null *BAP1* mutation had an earlier age of onset than patients with missense mutations. Both null and missense carriers displayed a lower age of onset for these tumors in comparison with the general US population. The authors also reported that among carriers of *BAP1* null mutations, peritoneal mesothelioma was more prevalent than pleural mesothelioma, in contrast to what was observed in the general population. No other associations were found.

### 4.7. What Can We Say about the TPDS Spectrum?

In this study, we performed IHC on tumor samples from patient MPM_HO1901, detecting loss of BAP1 expression in different tumor types. Of note, our NGS and IHC results on the patient’s LUAD tissue also suggest a possible functional role of *BAP1* in this tumor, albeit at a later stage of tumor progression. Interestingly, we observed loss of BAP1 expression in the nucleus and in cytosol of meningioma cells as well as in BCC cells from two different germline *BAP1* PV carriers belonging to two different families, confirming that meningioma and BCC are also included in the broader *BAP1*-TPDS spectrum [7,49,50].

### 4.8. New Pathogenic Mutation

In this study, we also describe three novel families, named PD-578, PD-238, and PD-601. Proband MM981 (PD-578) carries the p.Trp202* PV in *BAP1*, a previously unknown nonsense variant, and his melanoma shows loss of protein expression.

### 4.9. Survival of BAP1 Carriers

Since mean survival from diagnosis of malignant PlM (MPM) patients ranges between 9 and 17 months [51], five out of seven *BAP1* PV patients with PlM analyzed in this study should be considered as long-term survivors (Table 3). Indeed, patient II-1 (family A), who developed peritoneal mesothelioma when he was 63 years old (in 2001), lived for 72 months (6 years) after diagnosis. Similarly, proband II-5 (family A1) lived 108 months (nine years) after being diagnosed with PlM, while her brother died after 24 months. ID_5 III-1 also survived 24 months after the diagnosis. Interestingly, patient MPM_HO1901 (II-3) is still alive, 27 months after the diagnosis, and, since she is positive for a mutation in *BAP1*, has been recruited in a trial aiming to test the combination of immunotherapy and PARP inhibitors as second-line treatment (NCT04940637) [46]. These results agree with previous studies showing prolonged survival of PlM TPDS patients compared to that of sporadic patients [8,9,12,14,15,16,17,18].

## 5. Conclusions

We report here the results of our genetic screening of both germline and tumor samples from a cohort of *BAP1*-TPDS patients, identifying six different germline *BAP1* PVs in seven families. Overall, our findings stress the importance of an appropriate surveillance program for *BAP1*-TPDS carriers, which should involve not only those individuals who have already developed a tumor but also their healthy relatives carrying a *BAP1* PV.

Finally, we recommend that confirmation of biallelic loss in the tumor tissue should be carried out before recruiting patients for precision medicine-based clinical trials.

## Figures and Tables

**Figure 1 diagnostics-12-01710-f001:**
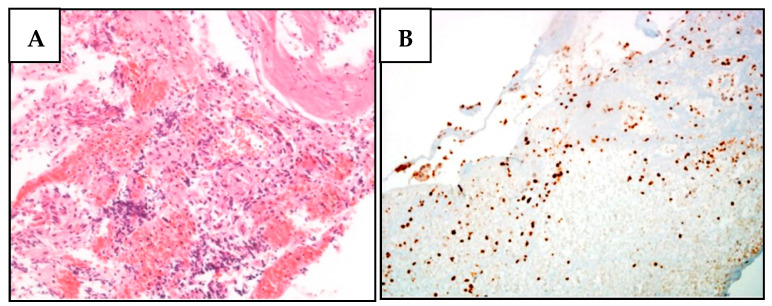
BAP1 nuclear expression is lost in meningioma cells of the proband’s daughter (III-2) (family A). (**A**) H&E staining of a relapsed meningioma section characterized by the presence of rare and minute fragments of a highly vascular round and spindle cell proliferation, with scant atypia and no evident mitoses. Tumor cells expressed EMA and smooth muscle actin, in the absence of synaptophysin, S100, CD34, and HMB45. (**B**) No BAP1 expression was observed in meningioma cells, while it was maintained in stromal cells.

**Figure 2 diagnostics-12-01710-f002:**
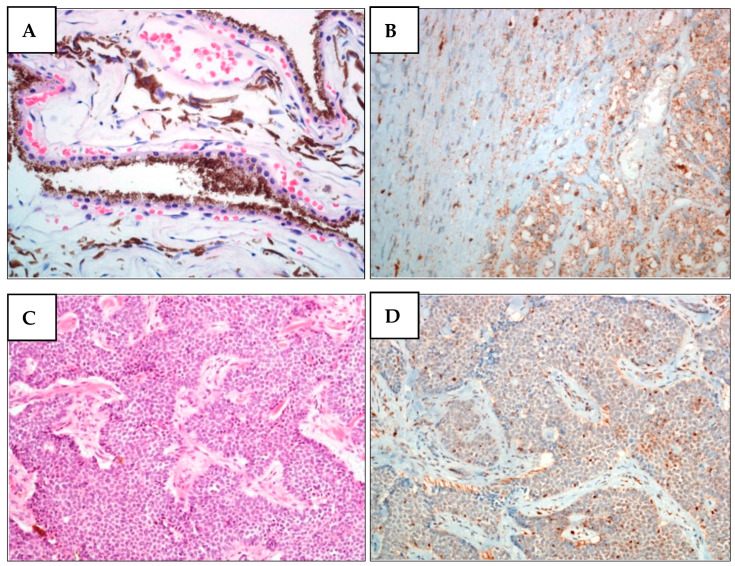
BAP1 nuclear expression is lost in melanoma cells from the proband’s brother (II-2) and in BCC cells from proband (II-5) (family A1). (**A**) H&E staining of II-2 UM. Unevenly pigmented epithelioid cell melanoma with radial extension involving over half of the iris and infiltrating the underlying connective tissue of the ciliary body. (**B**) BAP1 IHC of II-2 UM. The expression of BAP1 is lost in both nucleus and cytoplasm of tumor cells. (**C**) H&E staining of II-5 BCC. (**D**) BAP1 IHC of II-5 BCC. In basal cell carcinoma, BAP1 expression was only observed in lymphocytes and stromal cells but not in tumor cells.

**Figure 3 diagnostics-12-01710-f003:**
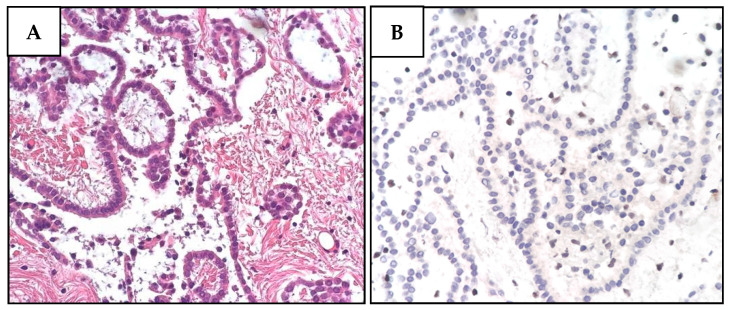
Histological and immunohistochemical features of proband MPM_HO1901 (II-3) tumors. (**A**) H&E staining of pleural mesothelioma (PlM), showing a diffuse, mostly superficial, tubular-papillary proliferation of well-differentiated neoplastic mesothelial cells with focal invasion of the adipose tissue. Neoplastic cells are positive for calretinin and WT1, while they are negative for TTF1 and CEA. (**B**) BAP1 immunohistochemical expression is lost in both the nucleus and cytoplasm of PlM cells, while it is retained in normal stromal cells. (**C**) Conventional LUAD reveals neoplastic glandular structures. The lesion is a well-differentiated adenocarcinoma with a predominant lepidic growth of neoplastic cells and focal areas of stromal invasion. (**D**) BAP1 expression is absent in adenocarcinoma cells, while it is retained in normal interalveolar histiocytes (arrow). (**E**) Renal cell carcinoma (RCC) showing tumor cells characterized by a distinct cell membrane and optically clear cytoplasm associated with more eosinophilic neoplastic cells. The grading score is moderately differentiated due to the presence of polymorphic nuclei, evident nucleoli, and rare mitoses. (**F**) BAP1 immunohistochemistry reveals a heterogeneous expression pattern including areas of nuclear retention (upper left) and areas with BAP1 nuclear and cytoplasmic loss (lower right).

**Figure 4 diagnostics-12-01710-f004:**
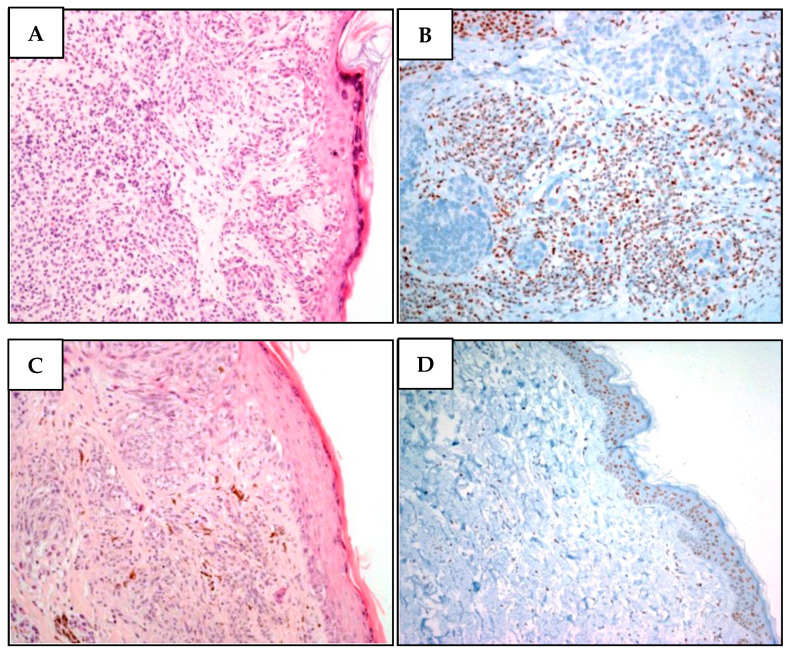
Two different melanomas of proband MM981. (**A**). Malignant melanoma cells showing superficial extension, vertical growth, absence of ulceration, and nevus residual with a bland lymphocytic infiltration. (**B**) While BAP1 is undetectable in neoplastic atypical melanoma cells, it is present in stromal cells. (**C**) Malignant melanoma section denoting superficial extension, vertical growth phase, absence of ulceration, with spindle cell appearance and poor lymphocyte infiltration. There are signs of regression in the absence of angiolymphatic and perineural invasion. (**D**) Lack of BAP1 immunohistochemical loss in atypical nevus cells, while the epidermal cells regularly express BAP1.

**Table 1 diagnostics-12-01710-t001:** Germline *BAP1* PVs identified in the families analyzed in the present study.

Family ID	Patient ID	*BAP1* Germline Variant	rs	MAF	Mutation	Localization	Tumor	Reference
A	A_II-1 §	c.46_47insA	-	-	Insertion	Exon 2	PeM	[35]
	A_I-2	c.46_47insA	-	-	Insertion	Exon 2	PlM	[35]
	A_III-1	c.46_47insA	-	-	Insertion	Exon 2	Mucoepidermoid carcinoma	[35]
	A_III-2	c.46_47insA	-	-	Insertion	Exon 2	Meningioma	This article
A1							CM	[31]
	A1_II-5 §	c.1153C>T p.Arg385*	rs1553645164	-	Stop gain	Exon 12	PlM	[31]
							BCC	This article
							Meningioma	This article
	A1_II-2	c.1153C>T p.Arg385*	rs1553645164	-	Stop gain	Exon 12	UM	This article
							PlM	This article
	A1_III-2	c.1153C>T p.Arg385*	rs1553645164	-	Stop gain	Exon 12	CM	This article
ID_5	ID_5 III-1 §	c.783+2T>C	rs774730309	<0.0001	Donor splice	IVS9	PlM	[32]
PD-601	MM1012 §	c.783+2T>C	rs774730309	<0.0001	Donor splice	IVS9	CM	This article
MPM_HO1901							PlM	[39]
	MPM_HO1901 II-3 §	c.38-1G>T	-	-	Acceptor splice	IVS1	RCC	[39]
							LUAD	[39]
	MPM_HO1901 II-2	c.38-1G>T	-	-	Acceptor splice	IVS1	PlM	This article
PD-578							CM	This article
	MM981 §	c.605G>A p.Trp202*	-	-	Stop gain	Exon 8	CM	This article
							Prostate cancer	This article
PD-238	MM400 §	c.376-2A>G	-		Acceptor splice	IVS5	UM	This article ^
							Bladder cancer	This article ^

MAF, minor allele frequency; LOH, loss of heterozygosity; PeM, peritoneal mesothelioma; PlM, pleural mesothelioma; CM, cutaneous melanoma; BCC, basal cell carcinoma; UM, uveal melanoma; IVS, intervening sequence; RCC, renal cell carcinoma; LUAD, lung adenocarcinoma; §, proband; ^, the variant was previously reported in an Australian family [7].

**Table 2 diagnostics-12-01710-t002:** Analyses on tumor specimens of the mutated patients and relatives.

Patient ID	Tumor	Tumor Type	IHC BAP1	LOH	CNVs	MLPA	FISH	Somatic Variant
A_II-1	PeM	NA	NA	NA	NA	NA	NA	/
A_I-2	PlM	FFPE	Not expressed	NA	NA	NA	NA	/
A_III-1	Mucoepidermoid carcinoma	FFPE	Not expressed	Yes	NA	NA	NA	/
A_III-2	Meningioma	FFPE	Not expressed	NA	NA	NA	NA	/
A1_II-5	CM	FFPE	Not expressed	No	NA	NA	NA	/
	PlM	FFPE	Not expressed	No	NA	NA	NA	/
	BCC	FFPE	Not expressed	NA	NA	NA	NA	/
	Meningioma	NA	NA	NA	NA	NA	NA	/
A1_II-2	CM	FFPE	Not expressed	NA	NA	NA	NA	/
A1_III-2	UM	FFPE	Not expressed	NA	NA	NA	NA	/
ID_5 III-1	PlM	FFPE	Not expressed	/	1.41 (*p* = 0.006)	NA	81% deletion ¤	*BAP1* c.783+2T>C (VAF 52.96%)
		Pleural effusion	/	Yes	/	50% deletion	38% deletion §	/
MM1012	CM	FFPE	Weak nuclear expression	Yes	NA	NA	NA	/
MPM_HO1901 II-3	PlM	FFPE	Not expressed	NA	0.96 (*p* = 0.00004)	NA	NA	*BAP1* c.38-1G>T (VAF 89.43%) *MSH6* p.Arg1076His (VAF 48%)
		Pleural effusion	/	Yes	/	50% deletion	NA	/
	RCC	FFPE	Heterogeneous expression ^#^	NA	/	NA	NA	*BAP1* c.38-1G>T (VAF 48.48%) *BAP1* p.Trp196* (VAF 2%) *MSH6* p.Arg1076His (VAF 48%)
	LUAD	FFPE	Not expressed	NA	/	NA	NA	*BAP1* c.38-1G>T (VAF 21.20%) *BAP1* p.Arg722His (VAF 4.23%) *BAP1* p.Arg718Gln (VAF 4.78%) *BAP1* p.Arg717Gln (VAF 9%)
MPM_HO1901 II-2	PlM	FFPE	Not evaluable ^#^	NA	/	NA	NA	*BAP1* c.38-1G>T (VAF 53.54%) *BAP1* p.Trp196* (VAF 3.36%) *MSH6* p.Arg1076His (VAF 25%)
MM981	CM	FFPE	Not expressed	Yes	NA	NA	NA	/
	CM	FFPE	Not expressed	No	NA	NA	NA	/
	Prostate cancer	FFPE	NA	NA	NA	NA	NA	/
MM400	UM	NA	NA	NA	NA	NA	NA	/
	Bladder cancer	NA	NA	NA	NA	NA	NA	/

IHC, immunohistochemistry; LOH, loss-of-heterozygosity; CNVs, copy number variants; MLPA, multiplex ligation-dependent probe amplification; FISH, fluorescent in situ hybridization; NA, not available; PeM, peritoneal mesothelioma; PlM, pleural mesothelioma; CM, cutaneous melanoma; UM, uveal melanoma; FFPE, formalin-fixed paraffin-embedded; VAF, variant allele frequency; BCC, basal cell carcinoma; RCC, renal cell carcinoma; LUAD, lung adenocarcinoma. ^#^ Uncertain interpretation possibly due to the sample condition; ¤ homozygous deletion 73% and heterozygous deletion 8%; **§** homozygous deletion 24% and heterozygous deletion 14% [32].

**Table 3 diagnostics-12-01710-t003:** Survival of mesothelioma patients carrying germline *BAP1* PVs.

Patient ID	Histotype	Age at Diagnosis	Survival
A_II-1	Epithelioid (Peritoneal)	63	72 months (6 years) *
A_I-2	Epithelioid (Pleural)	79	12 months
ID5_II-5	Epithelioid (Pleural)	53	108 months (9 years) *
ID5_II-2	Epithelioid (Pleural)	64	24 months *
ID_5 III-1	Epithelioid (Pleural)	52	24 months *
MPM_HO1901 II-3	Epithelioid (Pleural)	58	24 months *,°
MPM_HO1901 II-2	Sarcomatoid (Pleural)	32	4 months

*, long-term survivors; °, this patient is still alive.

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
