# Peer review of "Diagnostics of BAP1-Tumor Predisposition Syndrome by a Multitesting Approach: A Ten-Year-Long Experience"

_diagnostics, 2022, doi:10.3390/diagnostics12071710_

Round 1

Reviewer 1 Report

         Authors should indicate whether the family members presented in this study cover the whole families or there are immediate members of these families who have not been tested.

2      Please clearly indicate the novel observations regarding the studies of the families which have been reported elsewhere in the past and if possible, indicate how the previous and the new findings informed personal and clinical decisions. In addition, please indicate any potential action that could have been taken to avert the progression to disease.  

3      Please provide information if available on whether the mutations reported had any impact on the treatment and outcome of the disease in different family members.

4      Please indicate whether there is any link between the different types of BAP1 mutations and the stage of tumor or response to treatment.

5      Is there any prediction that could be drawn in terms of linking specific BAP1 mutations to specific disease based on gender, age, environmental or behavioral traits?

Author Response

Reviewer 1

[…]

Comments and Suggestions for Authors

1          Authors should indicate whether the family members presented in this study cover the whole families or there are immediate members of these families who have not been tested.

We thank the Reviewer for this observation. We have now specified in the text which family members were tested or not for each mutation as follows:

-  line 269: “..and four of them (i.e., the proband, her daughter and son, and the mother of the proband) carried the c.46_47insA PV (Figure S1). The other family members were not tested.”

- line 295: “…the nonsense c.1153C>T p.Arg385* variant (Figure S3). No other family members were available for BAP1 testing.”

- line 394: “Cascade genetic analyses were feasible for the proband’s brothers and daughters. One of…”

The pedigrees of the families in the Supplementary Materials offer a comprehensive perspective of the members that were tested – and resulted in carriers (+) or not (-) – or that have not been tested for the variant (no symbols).

2      Please clearly indicate the novel observations regarding the studies of the families which have been reported elsewhere in the past and if possible, indicate how the previous and the new findings informed personal and clinical decisions. In addition, please indicate any potential action that could have been taken to avert the progression to disease. 

We aimed to give a whole picture of both the past and the new analyses we performed, and we apologize if the novel data were not clearly identifiable. We have already indicated in the text if the observations regarding the studies of the families were reported in previous papers (i.e. Betti et al 2015-2016-2018 and Sculco et al 2022) or are novel data. The references are also included in Table 1 (p. 6, last column).

The results of BAP1 genetic testing are used by clinicians to select the suitable therapy for patients and to set up the surveillance program for family members (including genetic screening). For example, one of our patients, MPM-HO1901 – who carried a BAP1 pathogenic variant that we identified – was recruited for a clinical trial recently started in Turin (NCT04940637). We have now added this ClinicalTrials.gov Identifier in the text, specifying that she was eligible because of the PV (line 637-639). Moreover, cascade genetic analyses led to the identification of a healthy carrier (one of her daughters) for whom screening measures have been implemented (line 602).

3      Please provide information if available on whether the mutations reported had any impact on the treatment and outcome of the disease in different family members.

The identification of new BAP1 carriers has important implications for the proband’s family members such as the implementation of screening measures for a prompt diagnosis and treatments. We stress this concept in the discussion, please refer to: “4.2 The Importance of Detecting Secondary Somatic Mutations of BAP1 in Tumor Tissues”, “4.3 Which is the Best Strategy to Detect a Secondary Somatic Mutation?”, “4.6 Cascade Genetic Testing ” and “4.9 Survival of BAP1 Carriers”.

4      Please indicate whether there is any link between the different types of BAP1 mutations and the stage of tumor or response to treatment.

We thank the Reviewer for this comment. Given the low number of BAP1 carriers, correlation studies regarding different types of mutations and tumor stage or response to treatment are not possible.

5      Is there any prediction that could be drawn in terms of linking specific BAP1 mutations to specific disease based on gender, age, environmental or behavioral traits?

We thank the Reviewer for this observation. Unfortunately, we cannot make any prediction of our studied patients due to the low number of pathogenic variant carriers. However, in 2018 a comprehensive study collating data from 181 BAP1-TPDS families was published (Walpole et al. 2018). Interestingly, the Authors found that among patients that developed mesothelioma, melanoma and other tumors, those who carried a null BAP1 mutation had an earlier age of onset than patients with missense mutations. Both null and missense carriers displayed a lower age of onset for these tumors in comparison with the general US population. The Authors also reported that among carriers of BAP1 null mutations, peritoneal mesothelioma was more prevalent than pleural mesothelioma, in contrast to what observed in the general population.

It is also known that some BAP1-TPDS tumors are associated with environmental factors, for example UV light for cutaneous and uveal melanoma and asbestos for mesothelioma. We discussed this topic in “4.6 Cascade Genetic Testing” (Line: 582).

We have also added the following sentences:

- Line 593: “We have previously showed that the combined risk of genetic predisposition and asbestos exposure results in an increased risk of mesothelioma in the carriers of germline mutations [39]”.

Line 604: “Regarding associations between specific BAP1 mutations and a peculiar disease phenotype, in 2018 a comprehensive study [7] collating data from 181 BAP1-TPDS families showed that among patients who developed mesothelioma, melanoma and other tumors, those carrying a null BAP1 mutation had an earlier age of onset than patients with missense mutations. Both null and missense carriers displayed a lower age of onset for these tumors in comparison with the general US population. The Authors also reported that among carriers of BAP1 null mutations, peritoneal mesothelioma was more prevalent than pleural mesothelioma, in contrast to what observed in the general population. No other associations were found.”

Reviewer 2 Report

The authors present an excellent manuscript in an interesting area.  The article is extremely well written and well presented, with informative figures and tables, and a thoughtful and well-informed introduction and discussion.  The methods are well-documented, complete, and appropriate.

I find the conclusions to be perfectly valid and am convinced by the arguments for variant c.783+2T>C being deleterious and also the novel variant.

The only significant addition I would suggest would be to add a "lollipop plot" of all the variants discovered in the study.  There are webtools available to generate them, for example: https://joiningdata.com/lollipops/index.html

My other criticisms are incredibly minor:

Line 115 17.152 would clearer with a comma (i.e. 17,152)

Lines 479-482; this is placeholder text, please delete

In summary, congratulations and thanks for composing such an interesting and well-assembled article.

Author Response

Reviewer 2

[…]

Comments and Suggestions for Authors

The authors present an excellent manuscript in an interesting area.  The article is extremely well written and well presented, with informative figures and tables, and a thoughtful and well-informed introduction and discussion.  The methods are well-documented, complete, and appropriate.

I find the conclusions to be perfectly valid and am convinced by the arguments for variant c.783+2T>C being deleterious and also the novel variant.

The only significant addition I would suggest would be to add a "lollipop plot" of all the variants discovered in the study.  There are webtools available to generate them, for example: https://joiningdata.com/lollipops/index.html

We thank the Reviewer for this suggestion. We have now added a lollipop plot showing the variants we described in the present study mapped on the BAP1 gene. (line: 249, Figure S1)

My other criticisms are incredibly minor:

Line 115: 17.152 would clearer with a comma (i.e. 17,152)

Lines 479-482; this is placeholder text, please delete

We thank the reviewer for these suggestions; we have now modified the text.

In summary, congratulations and thanks for composing such an interesting and well-assembled article.
